# Crucial Role of Ni Point Defects and Sb Doping for Tailoring the Thermoelectric Properties of ZrNiSn Half-Heusler Alloy: An Ab Initio Study

**DOI:** 10.3390/ma17051061

**Published:** 2024-02-25

**Authors:** Eleonora Ascrizzi, Chiara Ribaldone, Silvia Casassa

**Affiliations:** Department of Chemistry, University of Torino, Via Giuria 5, 10125 Torino, Italy; eleonora.ascrizzi@unito.it (E.A.); chiara.ribaldone@unito.it (C.R.)

**Keywords:** thermoelectric materials, half-Heusler ZrNiSn alloys, density functional theory calculation, ab initio thermoelectric properties, Crystal code

## Abstract

In the wide group of thermoelectric compounds, the half-Heusler ZrNiSn alloy is one of the most promising materials thanks to its thermal stability and narrow band gap, which open it to the possibility of mid-temperature applications. A large variety of defects and doping can be introduced in the ZrNiSn crystalline structure, thus allowing researchers to tune the electronic band structure and enhance the thermoelectric performance. Within this picture, theoretical studies of the electronic properties of perfect and defective ZrNiSn structures can help with the comprehension of the relation between the topology of defects and the thermoelectric features. In this work, a half-Heusler ZrNiSn alloy is studied using different defective models by means of an accurate Density Functional Theory supercell approach. In particular, we decided to model the most common defects related to Ni, which are certainly present in the experimental samples, i.e., interstitial and antisite Ni and a substitutional defect consisting of the replacement of Sn with Sb atoms using concentrations of 3% and 6%. First of all, a comprehensive characterization of the one-electron properties is performed in order to gain deeper insight into the relationship between structural, topological and electronic properties. Then, the effects of the modeled defects on the band structure are analyzed, with particular attention paid to the region between the valence and the conduction bands, where the defective models introduce in-gap states with respect to the perfect ZrNiSn crystal. Finally, the electronic transport properties of perfect and defective structures are computed using semi-classical approximation in the framework of the Boltzmann transport theory as implemented in the Crystal code. The dependence obtained of the Seebeck coefficient and the power factor on the temperature and the carrier concentration shows reasonable agreement with respect to the experimental counterpart, allowing possible rationalization of the effect of the modeled defects on the thermoelectric performance of the synthesized samples. As a general conclusion, defect-free ZrNiSn crystal appears to be the best candidate for thermoelectric applications when compared to interstitial and antisite Ni defective models, and substitutional defects of Sn with Sb atoms (using concentrations of 3% and 6%) do not appreciably improve electronic transport properties.

## 1. Introduction

In recent years, thermoelectric (TE) materials have become of increasing interest thanks to electronic and transport properties that make them very suitable as green energy sources. In fact, by means of the Seebeck effect, they can transform heat given off from sources such as power plants, factories, motor vehicles or computers into electric power [1]. With their energy harvesting function, TE materials can contribute to face the climate crisis and transition towards a low-carbon economy [2,3].

Thermoelectric efficiency can be evaluated through the figure of merit ZT [4], which computed as $ZT=\sigma S^{2}T/\left(k_{e}+k_{l}\right)$, where σ represents the electrical conductivity, *S* is the Seebeck coefficient, *T* is the absolute temperature, and $k_{e}$ and $k_{l}$ are the electron and lattice contributions, respectively, to thermal conductivity. The goal in the TE field is to reach a high ZT value in order to make these systems attractive for practical applications. Optimizing ZT is not an easy task since an ideal TE should have a high power factor (PF), $\sigma S^{2}$, and a low value of $\left(k_{e}+k_{l}\right)$, while it is common for materials that have a high σ to also have high electron and lattice contributions to thermal conductivity [5]. Considering a temperature difference between hot and cold larger than 500 K in physical devices, values of ZT=1 and ZT=4 correspond to a conversion efficiency between heat and electricity that can reach percentages of 10% and 30%, respectively [6]. Materials with ZT=1 are presently commercially available [1]. However, increasing ZT by a factor of four, which would allow TE materials to be more suitable for practical purposes [7], has remained a formidable challenge.

Among different TE materials, half-Heusler (HH) alloys are one of the most studied for the mid-temperature regime: that is, for T≈(673,1073) K. This is the temperature range covered by common heat sources such as industrial waste heat and automobile exhaust [8]. These alloys (XNiSn and XCoSb with X = Hf, Zr, Ti) show excellent electronic transport properties and tunable band gaps [9], can be environmentally abundant, have low toxicity, and maintain chemical and mechanical stability at high temperatures [10]. Among HH materials, ZrNiSn is emerging due to its good thermal stability [11,12] and its high Seebeck coefficient that leads to promising values for PF and ZT [13].

Nevertheless, ZrNiSn-based HH alloys have not been extensively studied because their synthesis can prove difficult due to the complexity of their phase diagrams and the low melting point of Sn [12,14]. Moreover, experimentally, it is very challenging to surgically regulate the concentration and type of defects. In particular, it was demonstrated that a certain amount of Ni is soluble in the ZrNiSn matrix [12,15,16,17], and the presence of antisite Ni atoms is equally probable [17,18].

In this context, the possibility of modeling ZrNiSn and exploring the effects of local defects and low doping on its thermoelectric performance becomes particularly interesting and can complement the experimental results with an atomic-scale analysis of the structure–property relationship.

For these reasons, on the basis of Density Functional Theory calculations, we have investigated the pristine structure of ZrNiSn together with interstitial and antisite Ni defects by modeling through a supercell approach. The effect of doping the bulk structure with antimony atoms, which are known to increase both the carrier concentration and the electrical conductivity [19,20,21], has also been analyzed. A comparison with experimental data is carried out by exploring both the dependence on the temperature and the concentration of the charge carriers, allowing the various effects to be rationalized and providing an interpretative basis for the experimental data.

As already highlighted, ZrNiSn is a widely studied material: both theoretically and experimentally. However, we would like to stress some peculiarities of our work that can improve and enrich the present literature towards comprehension of the role of defects in ZrNiSn crystal. First of all, calculation of both the electronic band structure and the thermoelectric properties are performed using a single quantum mechanical package: the Crystal code. Crystal is capable of calculating the transport distribution function, together with the derived quantities, starting from the computed bands, which guarantees excellent reproducibility and internal consistency of the results. All these features implemented in the Crystal code were benchmarked in a first work of G. Sansone et al. [22] using demonstrative examples, including silicon and aluminum bulk crystals as well as thermoelectric materials such as CoSb_3_ and Bi_2_Te_3_, and in subsequent theoretical works [23,24]. Secondly, our study sheds light on the relationship between topological properties and electronic structure properties, thus allowing a glimpse of the role of bond characteristics for tuning the thermoelectric properties in ZrNiSn half-Heusler materials. Finally, to the best of our knowledge, this is the first theoretical study that manages to both reproduce the Seebeck behavior as a function of temperature (increasing or decreasing) and correlate it to the charge carrier concentration, which is one of the most difficult experimental parameter to measure and which is most affected by the synthesis path of the material itself.

The article is structured as follows: The computational setup is described in Section 2. In Section 3, results describing the structural and electronic properties of pristine and defective ZrNiSn models are presented, and the thermoelectric features of the different structures are analyzed in light of the most recent literature data. In particular, a comparison is carried out with the TiNiSn crystal, which is counted among the best performing half-Heusler alloys, to validate our methodology [24] and as a performance benchmark. Finally, in Section 4, the conclusions are outlined and, based on the results obtained, some suggestions are provided for improving the thermoelectric performance of ZrNiSn.

## 2. Computational Details

All the calculations are performed with the public version of the Crystal program [25,26]: an ab initio quantum chemistry package for solid state systems based on the Hartree–Fock method and Density Functional Theory (DFT). In Crystal, Bloch functions are expanded as a linear combination of atomic orbitals (AOs), and the basis sets consist of contracted Gaussian-type functions. In the framework of DFT, we adopted the gradient-corrected Perdew–Burke–Ernzerhof (PBE) functional [27]. The DFT exchange-correlation contribution is evaluated by means of numerical integration over the unit cell volume using a pruned grid [28,29] with the number of radial and angular points equal to 99 and 1454, respectively.

The basis set has to be chosen carefully in order to obtain an optimal description of the system since it influences all the electronic properties. In particular, for Ni and Sn atoms, the bases are those already tested and validated in a previous publication and consist of an all-electron basis set for Ni and a pseudo-potential one for Sn [24]. For Zr atoms, the all-electron basis set by Valenzano et al. [30] was modified by changing the last shell *d* exponent from the original value of 0.41 to 0.59. As for the Sb atom, the pseudo-potential basis set was taken from the work of Causà et al. [31].

Integration over the reciprocal space is carried out using a Monkhorst–Pack mesh with a shrinking parameter set to [6,6] and consisting of 145 k-points in the irreducible part of the first Brillouin zone for the perfect supercell, 68 k-points for the antisite structure, 32 k-points for the interstitial and 80 k-points for the Sb-doped supercell [32].

The Coulomb and exchange series, summed in direct space, are truncated using overlap criteria thresholds of [8,8,8]. Convergence for the self-consistent field algorithm is achieved up to a threshold of 10^−7^ Hartree on the total energy per unit cell [32].

Geometry optimization is performed using analytical gradients with respect to nuclear coordinates and unit cell parameters and a quasi-Newtonian scheme combined with Broyden–Fletcher–Goldfarb–Shanno (BFGS) Hessian updating [33,34]. The default convergence criteria are adopted for both gradient components and nuclear displacements.

The equilibrium bulk modulus *B*_0_; elastic constants *c*_11_, *c*_12_ and *c*_44_; Poisson ratio ν; and Young’s modulus *Y* are all calculated at the PBE level and obtained with Crystal by solving the third-order Birch–Murnaghan isothermal equation of state [35].

The topological analysis of the electron density ρ(r) is performed using the TOPOND code [36,37] according to the Quantum Theory of Atoms In Molecules And Crystals (QTAIMAC). This analysis provides the critical points (CPs) $r_{\mathrm{CP}}$ where the gradient of the density vanishes: $\nabla \rho (r_{\mathrm{CP}})=0$. CPs can be classified in terms of their type (r,s), where *r* is the rank and *s* is the signature. By referring to the Hessian matrix at the CP, the rank is given by the number of its non-zero eigenvalues, while the signature is the sum of the signs of the eigenvalues. Stable structures are characterized by all critical points of $\rho (r_{\mathrm{CP}})$ with rank 3. The CPs can be interpreted as chemically recognizable structures in a crystal: nuclei (3,−3), bonds (3,−1), rings (3,+1) and cages (3,+3).

Thermoelectric properties such as the Seebeck coefficient (*S*), electrical conductivity (σ), and electron contribution to the thermal conductivity ($k_{e}$) are computed by using the semi-classical Boltzmann transport equation theory and the frozen band approximation as implemented in the Crystal code [22]. A constant relaxation time approximation is also assumed, and a relaxation time equal to 18 fs is used for all the systems and temperatures considered following the study on ZrNiSn by Schrade et al. [38] that empirically obtained the values to reproduce experimental thermoelectric data. For calculation of the thermoelectric parameters, a denser mesh of 1000 k-points is used for the perfect system and the antisite, interstitial and Sb-doped supercells. The spin–orbit coupling effect is not considered in calculations, as this feature is, at present, not yet included in the public version of the Crystal code. Finally, in order to estimate the thermoelectric figure of merit ZT, the total thermal conductivity $k_{tot}=k_{e}+k_{l}$ is needed. The lattice contribution to the thermal conductivity $k_{l}$ cannot be calculated with Crystal, so we relied on the experimental measurements of $k_{tot}$ by Xie et al. [20], whose data are reported in Appendix A. With reference to this table, ZT values reported in Section 3.3 are computed at each temperature using the different values of $k_{tot}$ that were experimentally measured for the nominally perfect ZrNiSn and doped ZrNiSn_0.97_Sb_0.03_ as well as ZrNiSn_0.94_Sb_0.06_ samples.

## 3. Results and Discussion

### 3.1. Models and Structures

The crystal structure of cubic HH pure-phase ZrNiSn is assigned to the $F\bar{4}3m$ (216) space group and has an experimental lattice parameter of 6.11 Å at room temperature [39,40,41]. Similarly to other ternary HH alloys, atoms of Zr, Sn and Ni occupy fused face-centered cubic sublattices at Wyckoff positions 4a[0,0,0], 4b[1/2,1/2,1/2] and 4c[1/4,1/4,1/4], respectively [18].

Starting from the perfect ZrNiSn crystal, we investigated two of the most common point defects and two structures with different Sb vs. Sn dopant ratios. To simulate a defect concentration of ≈3%, a 2×2×2 supercell is designed, *SC96-P*, and contains a total number of 96 atoms (and 656 atomic orbitals). The antisite structure, referred to in the following as *SC96-A* and shown in the upper right panel of Figure 1, is created by moving a Ni atom to one of the empty full-Heusler positions. Then, to model an interstitial defect, named *SC97-I* and reported in the upper central panel of Figure 1, an atom of Ni is added to one of the empty full-Heusler positions to obtain a supercell of 97 atoms, corresponding to 1.03 Ni atoms for each Zr (and Sn) atom. This percentage is chosen considering that, experimentally, the composition of most alloys is found to be ZrNi_1+*x*_Sn, with x≈5% [16,20]. In particular, an interstitial defect of 3% leads to a measurement of ZT equal to 1.1 at a temperature of 873 K [17]. Finally, one or two atoms of Sn on 32 are replaced by Sb to obtain the ZrNiSn_1−*x*_Sb_*x*_ structures, *SC96-Sb*, with x=3% or x=6%, respectively, as drawn in the lower central and right panels of Figure 1. Modeling of these localized single defects represents the necessary first step to refine the computational apparatus and to be able to address the effects of multiple and combined defects in a more specific study.

With the aim of testing our methodology and calibrating the computational setup, we perform a complete structural characterization of the *SC96-P* model. The optimized structural parameter *a*; the equilibrium bulk modulus *B*_0_; the elastic constants *c*_11_, *c*_12_ and *c*_44_; its Poisson ratio ν and the Young’s modulus *Y* are all computed at the PBE level and are collected in Appendix A. The overall agreement with the experimental data is encouraging since, beyond the absolute values, the ratios between the magnitudes for the different coefficients is, in any case, preserved. As far as the lattice parameter is concerned, the computed value of 5.95 Ådiffers only by 3% from the experimental one of 6.11 Å [39]. This difference can be due to defects, the presence of which is recognized in the experimental sample [39].

Then, the defective models are addressed, and their structures are fully relaxed. In the *SC97-I* and *SC96-A* systems, the lattice parameter increases from the pristine value of 5.95 Å to 5.96 Å and 6.03 Å, respectively, whereas Sb-doping keeps the volume unchanged. In general, the introduction of defects does not lead to appreciable deformations to the lattice compared to the perfect crystalline structure, as evidenced by the very similar lattice parameters of the perfect and the defective models. This is consistent with the low formation energy calculated for these defects [42], which for our systems can be easily estimated only for the case of *SC96-A* to be ΔH=0.03 eV per atom by using the formula
(1)$$\Delta H=\left(E_{\mathrm{SC96-A}}-E_{\mathrm{SC96-P}}\right)/n_{at}$$
where $n_{at}=96$ is the number of atoms in the supercell, while *E**_SC96-P_* and *E**_SC96-A_* are the DFT ground state total energies of the optimized geometry at zero temperature for the perfect ZrNiSn and defective Ni antisite supercells, respectively.

Indeed, in the *SC97-I* and *SC96-A* models, the coordination pattern of the interstitial and antisite Ni atoms shows a peculiarity. As reported in Table 1, the first and third coordination spheres are unchanged, but the second is composed of six and five Ni atoms, respectively, at distances of 3.0 Å and 2.9 Å. This situation is similar to what occurs in the full-Heusler structure, where each Ni is surrounded by 6 Ni atoms at 3.2 Å. The consequent appearing of a specific Ni-Ni interaction can be envisaged and will be further investigated in the next section.

### 3.2. One-Electron Properties

The ground state electronic charges, computed using Mulliken and Hirshfeld methods, are reported for the different structures in Table 1. Despite the differences in the absolute values, the two methods reproduce the same scenario. In fairly good agreement with the literature results [2], in the perfect *SC96-P* crystal, the Zr atom is described as positively charged, while Ni and Sn show an increase in their charge density.

The corresponding electronic charge density map is reproduced in panels *a* and *d* of Figure 2. These plots, obtained as the difference with respect to the charge density of the isolated atoms, confirm a certain degree of charge transfer from Zr to the Ni and Sn atoms. In the *SC97-I* and *SC96-A* structures, the Ni in the defective position has, respectively, six and five nickel atoms at a distance of about 3 Åin a coordination almost identical to that which occurs in the full-Heusler compound, where each nickel atom has six nickel neighbors (see Table 1). In this case, the resulting effect is that of greater delocalization of the electron density, which produces a positively charged nickel atom. Similarly, interstitial and antisite nickels have a significantly lower negative charge than other nickels in the FH alloy.

In the doped system, the Sb atom is surrounded by six Ni atoms, and the interaction with this first coordination sphere is slightly stronger than for the substituted Sn. It can be stated that antimony, despite its excess charge, fits well into the ZrNiSn lattice and reveals a good tendency to interact with nickel.

In order to rationalize the effect of the local defects on the bonding framework, topological analysis of the electron density is performed and bond critical points (BCPs) are localized and characterized (see Table 2). In *SC96-P*, two BCPs are present between the Ni and Sn and the Ni and Zr atoms. Based on the values of the bond indicators in $r\equiv r_{\mathrm{BCP}}$—namely, the charge density ρ(r); the ratio between the potential and the positive kinetic energy densities |V(r)|/G(r); and the bond order, defined as the ratio between the total energy and the charge density H(r)/ρ(r)—both of these interactions can be classified as belonging to the so-called *transit* region, i.e., neither ionic nor covalent. As can be seen in Figure 3, where |V(r)|/G(r) is plotted as a function of the bond order, the Zr-Ni bond appears slightly more ionic than the Ni-Sn one, and this description is in close agreement with the one proposed by Fecher et al. [2] in their seminal paper. Turning to the defective structures, it can be seen that the values of ρ(r) for the atoms that do not interact directly with the defect are substantially the same as in the original alloy. In the case of the *SC96-A* system, the number of BCPs equals that of the perfect system, and the values of the different descriptors practically coincide. Interestingly enough, in *SC97-I*, BCPs appear between the interstitial Ni and its Ni neighbors at positions of about 3.0 Å, showing all the characteristics of incipient metallic interaction. These bonds lead to the formation of strongly localized electronic states within the forbidden region positioned just above the original system’s Fermi level. These bonds play a crucial role in imparting distinct quasi-metal features to the *SC97-I* band structure (see panel *b* of Figure 4).

Finally, based on the values of the two ratios |V|/G and H/ρ as shown in Figure 3, in the *SC96-Sb*-doped structures, the Ni-Sb bond is stronger than the Ni-Sn one, and it can be classified as a transit interaction with a more pronounced covalent character.

The electronic band structure has a crucial role in determining the thermoelectric parameters of materials; therefore, its accurate reproduction is fundamental to get reliable results about transport properties.

The unit cell of perfect ZrNiSn crystal has an indirect band gap of 0.51 eV, with the valence band maximum in Γ and the conduction band minimum at a high-symmetry X point with coordinates (0.5,0.0,0.5) in the primitive basis (see Appendix A). This band structure closely resembles the one obtained experimentally using angle-resolved photoemission spectroscopy [16]. To facilitate comparison with defective systems, the band structure of the perfect supercell, *SC96-P*, is shown in panel *a* of Figure 4. As highlighted by the density of the states of the *SC96-P* structure (Figure 4, panel *a*), the valence electronic states posses a strong Ni character combined with a relevant Zr *d*-orbital contribution. The overall description of the electronic band structure is in good agreement with literature data obtained at the PBE [21,44,45] and LDA [46] levels. However, to our knowledge, theoretical predictions tend to overestimate the experimental band gaps, which vary from 0.13 eV [39] to 0.18 eV [47], when the models do not sufficiently account for the presence of intrinsic defects. Indeed, on the one hand, Fu et al. [16] have recently synthesized a ZrNiSn crystal with a low concentration of defects, and depending on the technique used, they measured a band gap between 0.45 and 0.66 eV, which is very close to the one obtained for our *SC96-P* bulk model. On the other hand, Fiedler et al. [42] and Do et al. [48] have recently and independently computed such a correlation between local defects and the gap width. In this respect, our calculations show how both the interstitial and antisite Ni atoms introduce defective states in the energy region that is forbidden in the perfect system, causing a significant reduction in the main gap (see panels *b* and *c*, respectively, of Figure 4). On the one hand, the in-gap states due to interstitial and antisite nickel, which cross the Fermi surface and induce almost metallic behavior, could improve the electrical conductivity and therefore the power factor. On the other hand, narrowing of the gap could decrease the value of the Seebeck coefficient and thus worsen the thermoelectric performance. Calculation of the electronic transport properties emerges as crucial in this respect in order to evaluate the balance between these two contributions and the overall effect of Ni defects.

As regards the Sb-doped alloys, they retain the band structure of the pristine material with the addition of Sb states in the low-energy part of the valence region (see Figure 5) in agreement with previous literature results [21]. The extra charge due to the additional Sb electron with respect to the stoichiometric Sn is visible in the partial occupancy of the lower virtual states, as shown in Figure 5.

As a general comment, the presence of the defects in ZrNiSn in terms of stress exerted on the lattice structure introduces a negligible effect since the atomic distances and lattice parameters remain almost unchanged with respect to the perfect crystal. Conversely, even such a small percentage of site disorder and doping is capable of producing a sensitive change in the electronic structures near the Fermi level. This is consistent with the rather high percentage of this type of defect in ZrNiSn HH alloys (i.e., low defect formation energy) and with the marked effect they have on the electronic transport properties, which strongly depend on the band structure, as discussed in the next section.

### 3.3. Thermoelectric Properties

The thermoelectric features of pristine ZrNiSn are first calculated and compared with those of TiNiSn, which is the established benchmark among HH alloys [24,49,50,51,52,53]. This comparison, carried out on a common ground, allows us to evaluate the potential of ZrNiSn as a thermoelectric material. For the sake of clarity, the optimized cell parameters, electronic band structure and density of states of the crystalline phase of TiNiSn, as calculated at our level of theory, are reported in the Appendix A. The overall trend in the Seebeck coefficient in the two alloys is very similar, while the computed PF shows considerable differences. In particular, in the *n*-type region, ZrNiSn has a higher power factor with respect to TiNiSn (see Figure 6). Interest in this material, at least in its perfect crystalline form, is therefore reasonable.

From an experimental point of view, undoped ZrNiSn is unanimously considered an *n*-type material. However, experimental samples can present different concentrations of charge carriers $n_{c}$ depending on the synthesis route followed and have correspondingly different behaviors for the thermoelectric parameters as a function of temperature. In particular, recent studies in the literature have highlighted the existence of two types of materials: those that have a $n_{c}$ $\approx 10^{18}$ cm^−3^, for which the Seebeck coefficient *S* decreases with temperature *T* [39], and those with $n_{c}$ $\approx 10^{19}$ cm^−3^, for which the Seebeck coefficient increases with *T* [20,38].

Interestingly enough, our calculations are able to reproduce both behaviors. In Figure 7, the Seebeck coefficient at different temperatures is reported as a function of the carrier concentration. In the *n*-type region, for $n_{c}=2.8\cdot 10^{18}$ cm^−3^, the Seebeck coefficient *S* decreases with the temperature, while for $n_{c}=3.6\cdot 10^{19}$ cm^−3^, the opposite occurs. All the other transport properties, e.g., electron conductivity, σ, power factor, PF, electron contribution to the thermal conductivity and $\kappa_{e}$, are calculated at these two charge concentration carrier values at 300 K, and the results are collected in Appendix A. For example, for the case of $n_{c}=3.6\cdot 10^{19}$ cm^−3^, we get a PF $=14\cdot 10^{-4}$ WK^−2^m^−1^, which is similar to the value measured by Muta et al. [13] of $10\cdot 10^{-4}$ WK^−2^m^−1^, and the corresponding ZT is equal to 0.06, which is in perfect agreement with the experimental data of Shen et al. [19].

In Figure 8, the Seebeck coefficient of pristine ZrNiSn is reported as a function of temperature in the range (300,900) K, and the trend is compared with those obtained by Schmitt et al. [39] and Shen et al. [19] with experimental measurements of samples characterized by different values of $n_{c}$. The agreement is quite good, and the difference arises and becomes appreciable only in the high-temperature regime. As already noted by Shen et al. [19], the discrepancy can be attributed to the activation of electron–hole pair excitation across the energy gap at high temperature so that the opposite contributions from the two carriers reduce the absolute value of the observed Seebeck coefficient. This phenomenon cannot be reproduced by our theoretical calculations, which are based on the semi-classical model that treats the band index as a constant of motion and therefore does not consider interband transitions.

Moreover, an attempt is made to rationalize the effect of local defects on the thermoelectric performance of this alloy.

As regards the Ni-related defects, the Seebeck coefficient and the power factor of *SC97-I* and *SC96-A* are reported in Figure 9. At 300 K and $n_{c}$≈ 10^19^ cm^−3^, *S* decreases, and this reduction occurs to a greater extent in the case of *SC96-A*, as already observed by Fiedler et al. [42], and is probably due to the opposite contributions of the electrons in the conduction band and hole carriers in the in-gap states. The effect of the temperature increase is shown in Figure 9, where *S* and the electrical conductivity are reported in the range 300 K <T< 900 K: the general trends for the two properties in question are opposite and in good agreement with experimental results [41].

Finally, we studied the antimony-doped model, focusing on the percentage of Sb, which, experimentally, seems to be a crucial element. Indeed, on the one hand, independently of each other, the groups of Shen et al. [19] and Xie et al. [20] observed a decrease in the power factor PF as doping increased. On the other hand, the peculiarity of antimony, which is able to increase both the electron conductivity and the concentration of carriers, is well documented [16,19,20], and it is claimed to lead to a significant increase in PF [16,21]. Some researchers, such as Fiedler et al. [42], have argued that the quenching effect could be due to the presence of concomitant defects in the synthesized materials [42], and in this respect, the ability to simulate different concentrations of Sb in the absence of other types of defects can provide useful insights for a better understanding.

A chemical potential value of 0.58 eV, corresponding to a maximum PF (as highlighted in the inset of Figure 10), is chosen to calculate the figure of merit as a function of temperature for the 3% and 6% Sb-doped systems. For the perfect *SC96-P* model, this value corresponds to the usual concentration of carriers, i.e., $3.6\cdot 10^{19}$ cm^−3^, where, as expected, the number of carriers increases with doping and reaches values of $5.5\cdot 10^{19}$ cm^−3^ and $1.4\cdot 10^{20}$ cm^−3^, respectively, for the 3% and 6% Sb-doped models. The corresponding figures of merit, reported in Figure 10 and calculated using the experimental values of $k_{tot}$ measured by Xie et al. [20], show a trend as a function of temperature in good agreement with that found by Xie et al. [20]. In particular, (i) ZT increases with temperature for all models, (ii) the *SC96-P* system has the highest values for the whole temperature range, and (iii) an increase in the Sb concentration from 3% to 6% causes marked deterioration in the thermoelectric effectiveness. Based on our calculations, we can conclude that doping with the Sb atomic concentrations previously analyzed is not a good strategy to improve the electronic transport properties of ZrNiSn alloys.

## 4. Conclusions

By means of DFT-PBE calculations, we have provided a comprehensive study of the physico–chemical properties of bulk ZrNiSn half-Heusler alloy, with particular attention paid to the role of defects such as interstitial, antisite Ni atoms and substitutive doping with Sb on its thermoelectric properties.

Initially, we fully characterized the pristine and defective systems by computing their structural and one-electron properties (geometry relaxation, charge density and topological analysis according to Bader’s theory; band structure and density of states), obtaining good overall agreement with the theoretical and experimental data from the literature as well as some new information on the electronic structures of the defective materials.

In particular, topological analysis permits us to characterize the bonds between the interstitial Ni and its nickel neighbors as covalent interactions and to explain the presence of highly localized electronic states just below the bottom of the conduction band of the perfect system in the band structure of the interstitial defective model.

Leading by the quasi-metal features of the defective structures, which can possibly induce an increase in the conductivity and thus an enhancement of the transport properties, we then computed the thermoelectric observables within the frozen-band Boltzmann transport theory as implemented in the Crystal code. The Seebeck coefficients and the power factors of all structures were studied as a function of temperature and carrier density and were compared with those of the well-known benchmark material, TiNiSn, and with experimental results in the literature.

The presence of Ni interstitial and antisite defects leads to the expected increase in the electronic conductivity with respect to the perfect crystal. However, this enhancement is balanced by a decrease in the Seebeck coefficient over the entire range of temperatures explored by our calculations.

Regarding doping with Sb, in rather good agreement with observations of different experimental samples, the presumed potential due to the increase in the carrier density and the decrease in the thermal conduction results in no increases to performance at the considered doping concentrations. Therefore, the perfect ZrNiSn alloy has been demonstrated to be a very good candidate as a thermoelectric material, while the presence of local defects and Sb-doping (at the considered concentrations of 3% and 6%) are unable to introduce further improvement. At the same time, our computational method has been revealed to be capable of interfacing with experimental and theoretical data and can provide models and interpretations that are able to validate theses and descriptions of electronic transport phenomena with insight at the atomic level.

## Figures and Tables

**Figure 1 materials-17-01061-f001:**
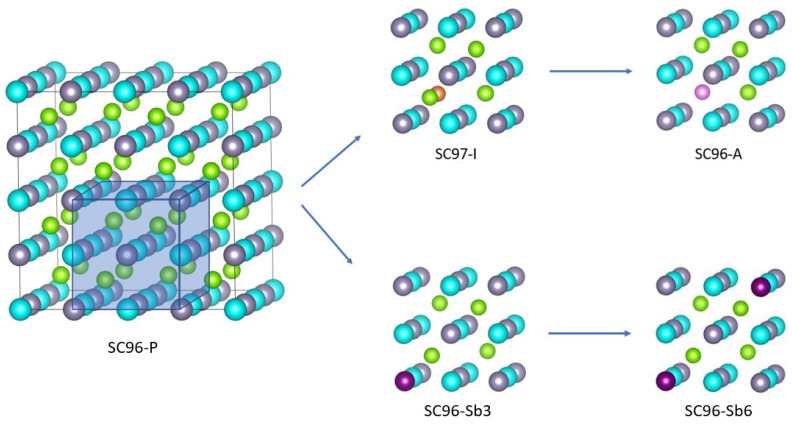
The supercell structures of ZrNiSn models: the bulk crystalline system *SC96-P* (**left** panel) together with its local modifications to get the *SC97-I*, *SC96-A* and *SC96-Sb* models (**right** panels). Color pattern for the atoms are: Zr, blue; Ni, green; interstitial Ni, orange; antisite Ni, pink; and Sn, gray.

**Figure 2 materials-17-01061-f002:**
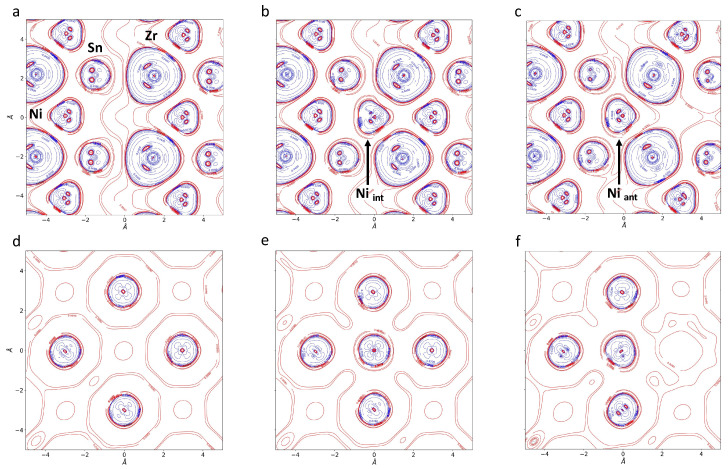
Electron charge density difference maps with respect to the atomic superposition, $\Delta \rho (r)=\rho_{\mathrm{cry}}\left(r\right)-\rho_{\mathrm{atoms}}\left(r\right)$, in the (101) plane (upper panels) and in the plane containing only Ni atoms (lower panels) for: *SC96-P* (**a**,**d**); *SC97-I* (**b**,**e**); *SC96-A* (**c**,**f**). Red and blue lines represent positive and negative values, respectively. The isolines correspond to the following values: ±0.08, 0.04, 0.02, 0.008, 0.004, 0.002, 0.0008, 0.0004 and 0.0002 electrons Bohr^−3^. This figure, as with the others regarding electron and transport properties, has been obtained by means of crystalpytools [43].

**Figure 3 materials-17-01061-f003:**
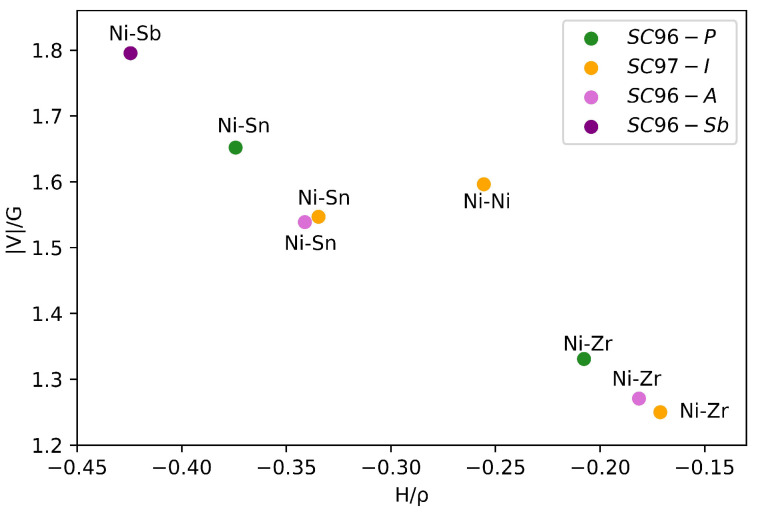
The ratio between the potential and positive kinetic energy densities |V(r)|/G(r) as a function of the bond order H(r)/ρ(r) for the perfect *SC96-P* supercell.

**Figure 4 materials-17-01061-f004:**
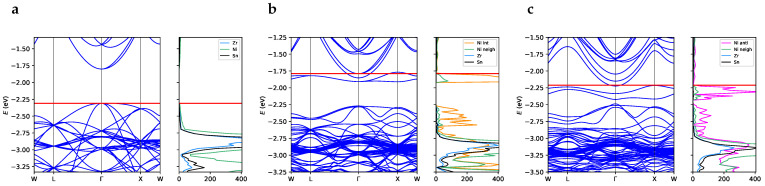
Band structure and projected density of states for (**a**) *SC96-P*, (**b**) *SC97-I* and (**c**) *SC96-A*. The red horizontal line represents the Fermi level. The band width for the in-gap states is 0.14 eV for *SC97-I* and 0.21 eV for *SC96-A*. Note that the labels of the reciprocal space points refer to the same values of the vectorial components used for the primitive unit cell of the perfect structure (Appendix A) but are applied to the reciprocal lattice vectors of the corresponding supercell.

**Figure 5 materials-17-01061-f005:**
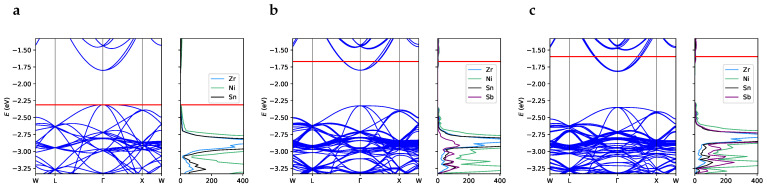
Band structure and projected density of states for (**a**) *SC96-P*, (**b**) *SC96-Sb* 3% and (**c**) *SC96-Sb* 6%. The red horizontal line represents the Fermi level. Note that the labels of the reciprocal space points refer to the same values of the vectorial components used for the primitive unit cell of the perfect structure (Appendix A) but are applied to the reciprocal lattice vectors of the corresponding supercell.

**Figure 6 materials-17-01061-f006:**
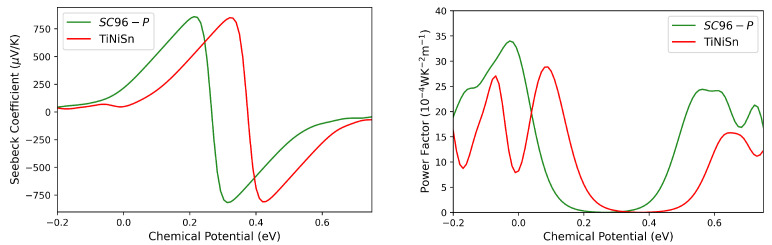
Seebeck coefficient (**left** panel) and power factor (**right** panel) as a function of chemical potential for *SC96-P* (green lines) and TiNiSn (red lines) at 300 K. The Fermi level corresponds to the zero value of the chemical potential.

**Figure 7 materials-17-01061-f007:**
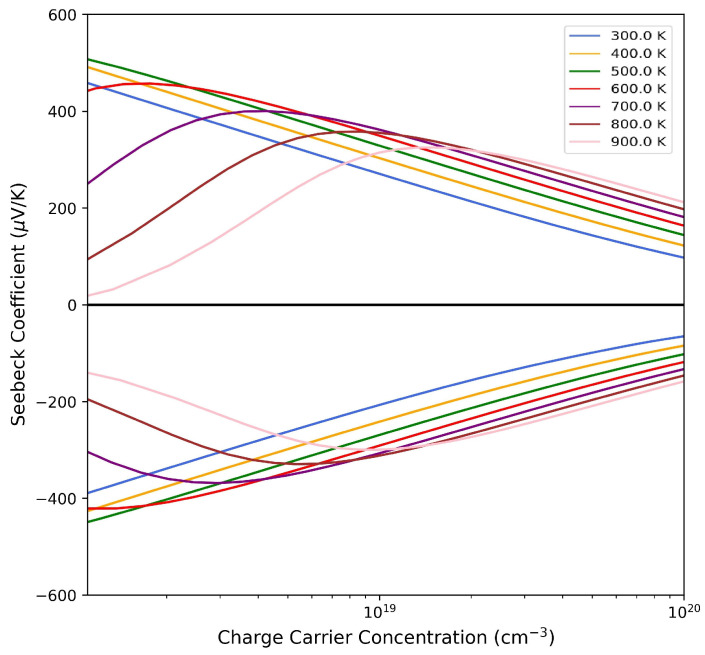
Seebeck coefficient as a function of the carrier concentration $n_{c}$ for *SC96-P* at different temperatures for *p*-type (**upper** panel) and *n*-type (**lower** panel) conduction.

**Figure 8 materials-17-01061-f008:**
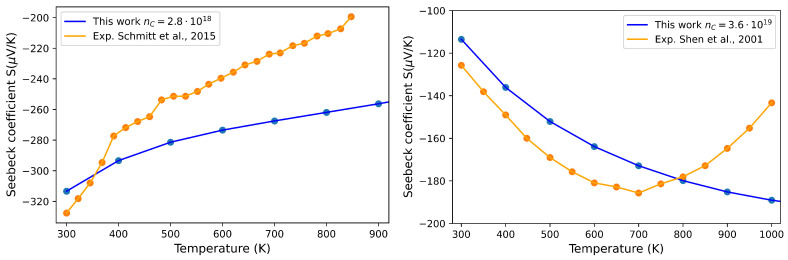
Seebeck coefficient as a function of temperature for different carrier concentrations in the perfect ZrNiSn crystal: $n_{c}\approx 3\cdot 10^{18}$ cm^−3^ (**left** panel) and $n_{c}\approx 3\cdot 10^{19}$ (**right** panel). Blue dots represent the values computed in this work, while orange dots are the experimental data from Schmitt et al. [39] (**left** panel) and Shen et al. [19] (**right** panel).

**Figure 9 materials-17-01061-f009:**
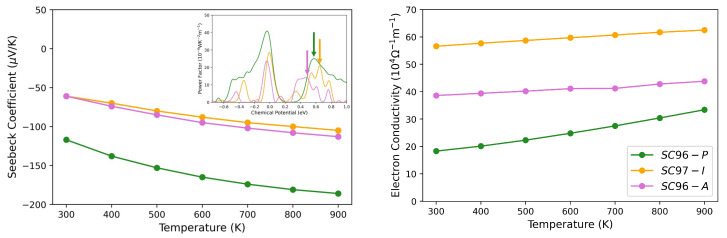
Seebeck coefficient (**left**) and electron conductivity (**right**) as a function of temperature for *SC96-P* (green lines), *SC97-I* (orange lines) and *SC96-A* (pink lines). In the inset, PF as a function of chemical potential is reported, and the arrows point to the values of PF at which *S* and σ are evaluated, which correspond to $n_{c}$ equals $3.6\cdot 10^{19}$ cm^−3^, $1.4\cdot 10^{20}$ cm^−3^ and $1.1\cdot 10^{20}$ cm^−3^ for *SC96-P*, *SC97-I* and *SC96-A*, respectively, at 300 K.

**Figure 10 materials-17-01061-f010:**
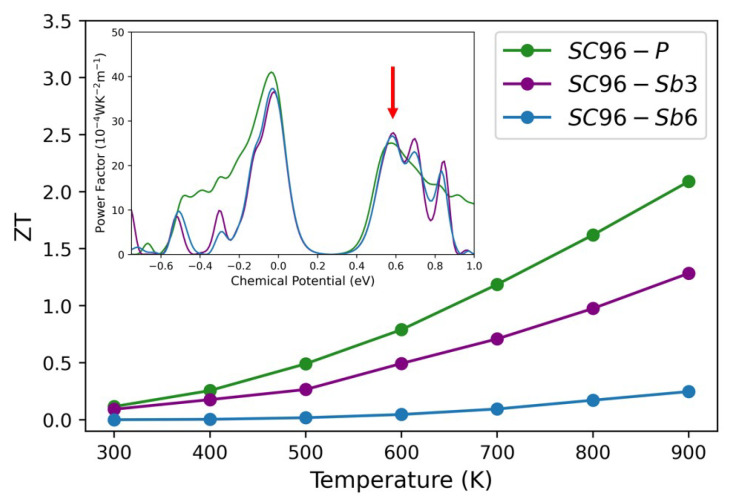
The figure of merit ZT as a function of temperature for *SC96-P* (green line), *SC96-Sb* 3% (purple line) and *SC96-Sb* 6% (blue line). Insert: power factor for the same structures as a function of chemical potential at 300 K. Here, the red arrow points to μ=0.58 eV, where all the ZT values have been evaluated considering the empirical relaxation time of 18 fs [38] and the experimental values of $k_{tot}$ obtained in Ref. [20] and reported for completeness in Appendix A.

**Table 1 materials-17-01061-t001:** Mulliken and Hirshfeld charges (a.u.) and coordination spheres (distance in Å) for the defective interstitial and antisite Ni atoms (indicated, respectively, as Ni_*int*_ and Ni_*ant*_) in *SC97-I* and *SC96-A* structures. For sake of comparison, the data for the full-Heusler structure, obtained using a basis set optimized for the metallic system as referred to in the Appendix A, are also provided.

Structure	Atom	Charge		Coordination	
		Mulliken	Hirshfeld	Distance	Neighbors
*SC96-P*	Ni	−0.981	−1.262	2.6	4 Zr, 4 Sn
				4.3	12 Ni
	Zr	+2.743	+2.746		
	Sn	−1.763	−1.501	2.6	4 Ni
				3.0	6 Zr
*SC97-I*	Ni_*int*_	−0.679	−0.980	2.6	4 Zr, 4 Sn
				3.0	6 Ni
				5.2	8 Ni
*SC96-A*	Ni_*ant*_	−0.643	−0.943	2.6	4 Zr, 4 Sn
				2.9	5 Ni
				5.2	8 Ni
*Full-Heusler*	Ni	+0.380	+0.455	2.7	4 Zr, 4 Sn
				3.2	6 Ni
	Zr	+0.707	+0.402		
	Sn	−1.467	−1.313		
*SC96-Sb*	Ni	−1.050	−1.281		
	Zr	+2.734	+2.714		
	Sb	−1.390	−0.957	2.5	6 Ni
				3.0	8 Zr

**Table 2 materials-17-01061-t002:** Bond critical points (BCPs) through topological analysis of the electron density. Bond length $b_{l}$ and distance of BCP from the Ni atoms $d_{\mathrm{Ni}}$ are in Å. The electron density ρ (e/Å^3^), its Laplacian $\nabla^{2}\rho$ (e/Å^5^), the bond degree H/ρ (Ha/(e/Å^3^)), the ratio between potential and kinetic density |V|/G, and the Hessian eigenvalue $\lambda_{3}$ (e/Å^5^) are evaluated at the BCP. Because $\lambda_{1}=\lambda_{2}$, the ellipticity $\epsilon =\lambda_{1}/\lambda_{2}-1$ is equal to zero.

System	Bond	$b_{l}$	$d_{\mathrm{Ni}}$	ρ	$\nabla^{2}\rho$	$\lambda_{3}$	H/ρ	|V|/G
*SC96-P*	Ni-Sn	2.579	1.186	0.058	0.046	0.091	−0.3742	1.6520
	Ni-Zr	2.579	1.276	0.050	0.084	0.166	−0.2077	1.3310
*SC97-I*	Ni*_int_*-Sn	2.645	1.233	0.047	0.052	0.084	−0.3346	1.5468
	Ni*_int_*-Zr	2.645	1.288	0.043	0.088	0.015	−0.1712	1.2501
	Ni*_int_*-Ni	3.055	1.444	0.036	0.025	0.043	−0.2556	1.5963
*SC96-A*	Ni*_ant_*-Sn	2.600	1.212	0.050	0.058	0.095	−0.3411	1.5388
	Ni*_ant_*-Zr	2.610	1.281	0.045	0.087	0.163	−0.1814	1.2709
*SC96-Sb*	Ni-Sb	2.530	1.164	0.063	0.027	0.082	−0.4245	1.7953

## Data Availability

The data presented in this study are available on request from the corresponding author.

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
