# Peer review of "Crucial Role of Ni Point Defects and Sb Doping for Tailoring the Thermoelectric Properties of ZrNiSn Half-Heusler Alloy: An Ab Initio Study"

_materials, 2024, doi:10.3390/ma17051061_

Round 1
Reviewer 1 Report
Comments and Suggestions for Authors
ZrNiSn is a widely studied material, thus a significant effort is needed to report anything new. The only novelty in this paper is the use of the Crystal code (but not benchmarked), and the topological bond analysis. None of the results are new, and the interpretation of results are dubious.
In some cases, a validation study of previous work can be in order, more of a literature study, demanding extensive literature review and comparison of specific finding of other authors. This is also missing. There are numerous studies of thermoelectric properties of ZrNiSn, and there are numerous studies of the doping mechanisms.
The conclusion that Sb-doping does not improve properties is completely unfounded, given that the authors do not propose any other way of doping the material. How to shift the Fermi level without dopants? This must go hand in hand with analysis of the scattering.
The analysis of the effect of Ni-defects on thermoelectric properties is of limited value without an account of how this affects the electron scattering. Moreover, there is no proper analysis of how it for instance impact the band structure, which might be helpful in this regard.
The energetic position of impurity states, still a research question of some debate, is not analyzed.
It is also worth noting that there are no DFT codes to compute lattice thermal conductivity. It is only a question of interfacing CRYSTAL to phono3py for instance. So this statement is misleading.
I also find it surprising, in the introduction, that the authors do not think TE materials currently are suitable for practical purposes.
Comments on the Quality of English LanguageSome room for improvement.
Author Response
Please see the attachement

Reviewer 2 Report
Comments and Suggestions for Authors
The author had performed calculations for Ni-points defects and Sb-doping for ZrNiSn half-Heusler alloy to study the thermoelectric properties using the DFT method. Overall, this paper presents very detailed results for perfect and defect structures and provides comprehensive analysis regarding their results. Additionally, The authors have compared results with peer research and see very good agreement with theoretical calculations, experimental data, etc. This cross validation makes the study in the paper in high transparency and a convincing way. The paper should be published.
I wonder if the authors any insights to offer regarding two questions:
-
As in the end of computational details, The author explains that experimental k_l cannot be calculated with CRYSTAL, so experimental k_tot is referenced from paper by Xie et al. In the supporting material TABLE S2, there is only ZrNiSn half Heusler alloy results. I understand the authors could reverse engineering to get the k_total or k_l from there. However, there is no defect structure k_tot or k_l info, does the author assume it doesn’t change with doping? In addition, how about the k_l change with temperature? I wonder if the author may provide any insights/explanation?
-
Regarding the computational details the author described for the basis used I wonder why in particular AE basis is used for Zr, while other atoms sounds are not? Does the calculation apply ECP/PP (Effective Core Potential/Pseudopotential)? If so, why in particular such a mixture of basis sets are chosen?
Additional are minor changes need improvement:
-
Keywords are now empty and need to be added.
-
All citations are broken, and need to be fixed.
-
In figure 4 and 5, the labeling of sub-figures (a), (b), (c) are not fitted to the corresponding figures
-
In Figure 9, not left and right as described in the figure title, but upper and lower
Reviewer 3 Report
Comments and Suggestions for Authors
In this manuscript, the authors performed DFT calculations on the half-Heusler ZrNiSn. In the frame of the Boltzmann transport theory as implemented in the CRYSTAL code, it is computed the values of some electron transport parameters, as the Seebeck coefficient and the power factors. Their dependence on temperature and carrier concentration is envisaged and the good agreement with the experimental counterpart allow us a possible rationalization of the effect of the various defects on the thermoelectric performances of the synthesized samples. It shows the defect-free material appears to be the best candidate for thermoelectric applications and doping with Sb does not appreciably improve its properties.
This study can be helpful for the computational chemistry and the industrial community. As such, the proposed article deserves to be published and the Materials is certainly well targeted.
Before the publication, I would like to ask the authors to consider the minor comments below.
1. page 2, line 85
“we adopted the gradient-corrected Perdew-Burke-Ernzerhof (PBE) functional”
For the systems studied in this work, the long-range interaction can be important. Can the authors discuss the effects of not adding dispersion correction to the PBE functional in DFT calculations?
2. page 7, Figure 4
Adding the band width results for the metallic systems in this figure can be helpful to illustrate the metal behaviors.
3. page 8, Figure 5
The DOS of conduction bands are very small compared with the dos of valence bands, can the authors provide a brief explanation for this issue?
4. page 10, Figure 10
In the subfigure on the right hand side, the trend of the calculated Seebeck coefficient has a different trend compared to the experiment results, can the authors provide more discussions on this?
5. page 2, Computational Details
Did the authors use finite temperature smearing in DFT calculations? If yes, it should be specified in computational details.
Comments on the Quality of English LanguageNo major language or grammar problem found.
Reviewer 4 Report
Comments and Suggestions for Authors
In their study, authors attempt to reproduce experimental data using DFT and Boltzman equation as implemented in CRYSTALS code aiming at understanding impact on the thermoelectric properties of ZrNiSn of interstitial and anti-site Ni, and the replacement of Sn with Sb in the structure. They benchmark their theoretical approach against the well-known TiNiSn system. While computations and comparison with the experimental results are agreeable, I do have some objections. First of all, authors should work on their English to make it more readable, starting with Abstract. While some paragraphs are nicely written, others are lacking in clarity. There are also some typos that should be corrected, I tried to mark some.
The main objection to the authors’ approach is the way that they model their crystals, i.e. defects are perfectly periodic in 2x2x2 supercell. In the resulting crystal structure, there is no disorder in the defects whatsoever. They don’t even mention the possibility of putting their defect onto different position in the supercell. I do understand that extending crystal lattice even further has heavy toll on the computation, but the analysis as it is done here does not inspire confidence in obtained results, thus considerably weakening authors’ conclusions. Before accepting the manuscript for publication, authors should address this issue in more detail.
In the row 167, authors claim that “no appreciable tensions are observed in the lattices”. If crystal structure is relaxed within DFT code, no considerable tensions should be present, so I do not see the purpose of this sentence. Furthermore, in accordance with the above comment, the lattice is perfectly periodic, and this further reduces the possibility of tensions. Please address this in more detail.
In row 187, Fig. 3 is referenced before Fig 2 that appears only in row 211. Also, While all figures are referenced in text as abbreviated Fig. xy, only figure 2 appears as Figure 2. – this should be made uniform throughout the text.
In the row 209, there is character æ that should probably be the Greek letter rho. Please check and correct. BTW, this is not the only place in the text that this occurs.
In the row 224, I am struggling to understand authors’ English.
Figure 3 should be positioned away from the edge of the paper. Also, there is no legend/scale for the figure, so it is impossible to read it.
Figure 4 is also messed up, panel designations are lost, and panels messed up, please correct this to make figure readable.
Font on the scales and legends in all figures must be increased on some figures to be easily readable and should be uniformized throughout the text on all figures.
Tables – fonts are bigger than in the main text. Should it be changed?
Figure 5 - same as figure 4 – arrange panels correctly, and fix scales and legend font size.
Row 294 – authors claim to present only “the results, whose agreement with literature data is encouraging” – I would urge the authors to present all their data, otherwise it looks as if they are deliberately presenting their results better than they really are. Scientific work should be honest and transparent. Also, authors should discuss discrepancies with other works, especially theoretical ones (since this is also theoretical study), and more so in the cases where order of magnitude discrepancy is observed. Now it seems that theoretically, one can calculate whatever and then take those results that match experiment without any new insight. Some conclusions must be made, otherwise, what is the contribution of this study, when all code and models are developed by other researchers.

Comments on the Quality of English LanguageAuthors should work on their English to make it more readable, starting with Abstract. While some paragraphs are nicely written, others are lacking in clarity. There are also some typos that should be corrected, I tried to mark some.
English:
Row 18 – phrase at the beginning, and after the coma
In the row 198 I do not understand the meaning of the word “countourned”
Row 355 – should it be Nickel instead of “nichel”?
Round 2
Reviewer 4 Report
Comments and Suggestions for Authors
In their response, authors appropriately addressed most of my comments. However, to have this published I would still suggest some changes.
1. Regarding my main objection to the way the defects were modeled. I do understand that theoretical calculations require periodic structures. But improvements can be made at least by doubling or tripling the cell and having two defects at different positions. Current approaches is minimalistic and I have serious doubts about it. If authors do not have technical means to do this (since calculations can become progressively more expensive), at least a note in the text should be put and this point needs to be discussed at the appropriate place in the text. Authors should honestly stress the limitations of their model and approach.
2. Regarding remark 7 and the new Figure 2, I would still prefer to have a legend. The numbers on the isolines in the figure are quite small and overlapping, making them unreadable even when fully zoomed in. Moreover, the numbers that can be discerned have no correlation with the scale (-8,8) mentioned in the caption. Therefore, why not place the legend next to the figure to enhance readability?
3. Regarding Figure 4, now that I can see it clearly and correlate it with the text, I notice that panels b and c contain more bands than panel a, even though the unit cells for all panels are the same. I understand that this band "multiplication" results from the degeneracy breaking upon the introduction of the impurity. However, for a more realistic representation, the authors should compare unfolded spectra into the large Brillouin zone of the small 1x1x1 unit cell. Currently, all bands are represented with the same weight, but the true impact of the impurity can be appreciated only when examining the unfolded spectra. Moreover, these spectra should have the closest resemblance to those observed experimentally - using angle-resolved photoemission spectroscopy. The same consideration applies to Fig 5.
4. What happened to the references?

Comments on the Quality of English Language5. Regarding English language: In the present era of AI, where English language and style polishing are accessible to everyone, it is unacceptable to me that readers should struggle with the readability of papers. The sentences in the text tend to be overly lengthy, and the word order appears awkward and inappropriate. This often necessitates readers to go through sentences multiple times to grasp the intended meaning while sometimes inevitably missing it. If a message is not conveyed in short, concise sentences, it can easily be lost. As a non-native English speaker, I find this even more challenging. Fortunately, a solution is available to everyone today, and it doesn't require much effort. So please invest some more effort in your manuscript.
